# Future Work and Disability: Promoting Job Motivation in Special Employment Centers in Spain

**DOI:** 10.3390/ijerph16081447

**Published:** 2019-04-24

**Authors:** Marina Romeo, Montserrat Yepes-Baldó

**Affiliations:** Social Psychology and Quantitative Psychology, University of Barcelona, 08035 Barcelona, Spain; myepes@ub.edu

**Keywords:** motivation, self-efficacy, responsibility for outcomes, meaningfulness of work, knowledge of results, special employment centers, employees with disabilities

## Abstract

The technological transition currently taking place in the labor market is having severe implications for people. One vulnerable group at risk of marketplace exclusion are employees with disabilities. This research explores their job motivation, including the moderated effect of self-efficacy. A 187 employees from Special Employment Centers (SECs) in Spain with intellectual and physical disabilities completed the Internal Motivation Scale, the Psychological Critical States (PCS) and the self-efficacy sub-scale of the Psychological Processes Scale (PPS) tests. Following the International Tests Commission guidelines, the instruments were adapted to the special needs of the participants. We found differences depending on the kind of disability of employees. In employees with intellectual disabilities, their levels of self-efficacy moderated the effect of responsibility for outcomes and meaningfulness of work on motivation. In the case of employees with physical disabilities, the knowledge of results is a predictor of motivation when they had medium or high levels of self-efficacy. Additionally, in this group, responsibility for outcomes and meaningfulness of work had a direct effect on motivation, independently of their levels of self-efficacy. Employees with disabilities should be empowered to make choices and exercise control in their working lives. To do so, human resources managers should promote their wellbeing, taking into consideration the workforce diversity.

## 1. Introduction

The technological transition currently taking place in the labor market is having severe implications for people’s wellbeing and relations with their jobs. Specifically, some authors and institutions [1,2,3,4] have pointed out that artificial intelligence, automation and robotics will create new jobs and destroy others. In this context, there exist some vulnerable groups at major risk of marketplace exclusion. One of these groups are employees with disabilities as stated by the United Nations in the Sustainable Development Goals [5]. The legal definition on disability in Spain is regulated by the Royal Decree 1971/1999, 23 December, Procedure for the recognition, declaration and qualification of the degree of disability (Spanish Law) [6]. Following this law, a person should have a minimum of 33% of limitation on his/her activities of daily living and/or social integration.

The United Nations Development Programme (UNDP) pointed out: “disability has a significant economic and social impact on persons with disabilities and their families, as well as on their communities and society. Employment rates are lower for men and women with disabilities than their peers without disabilities and excluding persons with disabilities from the world of work can cost countries between 1% and 7% of Gross Domestic Product. Persons with disabilities are at an increased risk of poverty due to this reduced access to employment” [7] (p. 9). According to the Public Service of Employment (SEPE) [8], in Spain, among the working age population, 6.12% have a legally recognized disability, but only 35.16% are active (working or seeking a job) and the unemployment rate is 9 points above the rate for people without a disability (22.62% versus 19.49%). Finally, it is important to note that most contracts are done on manufacturing, cleaning, and other occupations in the services sector (81.71%).

Some countries protect the jobs of these employees by developing sheltered work environments. In Spain, since 1985, the law defines these sheltered places as Special Employment Centers (SECs). The SECs are defined as organizations that should have as a social objective the inclusion of people with disabilities and can be created directly by the public administrations or by natural or legal persons who meet the appropriate civil requirements [9]. Following the current regulations, the SECs are required to have “the necessary instruments to be able to have structures, systems, processes and forms of organization that guarantee their competitiveness, not only to maintain themselves regularly in the market, but also to increase their position in it” ([10], p. 24).

Special Employment Centers (SECs) have been clearly reinforced during the crisis years, with a change in the hiring dynamics. In this sense, in 2006 there was a certain parity between the contracting developed by ordinary companies (52.5%) with respect to that carried out by the SECs (47.5%). However, as the unemployment rate increased, the hiring of persons with disabilities gradually declined in the ordinary market and increased under the protected employment umbrella. In the last two years there has been a decline in the general unemployment rate, reflected in a slight increase in contracts in ordinary companies (29.4%), although the percentage of contracts in SECs has not been affected (70.6%) [11]. The most common activities developed by SECs occur in the services sector (administrative activities, services to buildings and gardening activities, and Public Administration).

In this context, it is essential, therefore, to consider the employee with disabilities as a full-fledged employee, being necessary “to attract and retain qualified employees, many of whom will experience disability during their working years” [12] (p. 101).

It is important to remark the scarce empirical evidence in scientific literature focused on the collective of employees with disabilities. Nevertheless, recently this collective has gained more attention in the field of human resources, as pointed out by Addabbo et al. [13], and several authors has been addressed to analyze psychosocial characteristics such as job satisfaction, quality of work life, work motivation [14,15], motivation for the job [15,16], or motivation for employment in general [17].

Wanberg et al. [18] point out that motivation for the job has been found to be a predictor of employment success in the general population, and Andrews and Rose [17] noted that it is also a factor in employment for people with intellectual disability (ID). Specifically, Rose et al. [19] confirmed the findings of other authors, in the sense that “Motivation is a factor that may be influential in determining whether people with learning disability gain employment. It certainly seems to be a factor that influences whether individuals stay in their jobs for 3 months or more” [20] (p. 179).

Three internal psychological states are needed for work motivation: experienced meaningfulness of the work, experienced responsibility for work outcomes, and a knowledge of results [21,22,23,24]. According to the authors, experienced meaningfulness of work is “the degree to which the individual experiences the job as one which is generally meaningful, valuable, and worthwhile” ([21], p. 256); experienced responsibility for work outcomes refers to “the degree to which the individuals feels personally accountable and responsible for the results of the work he or she does” ([21], p. 256); and knowledge of results is defined as “the degree to which the individual knows and understands, on a continuous basis, how effectively he or she is performing the job” [1] (p. 257).

Quijano and Navarro [25] have considered the need to establish a “motivation equation”, because “it allows integrating various components of this phenomenon, in order to establish and measure the level of effort that people are willing to carry out and maintain in their work, as well as orienting roads and forms of intervention when the levels of motivation reached are low” ([25], p. 340).

According to these same authors, one of the most significant constructs for the analysis and evaluation of work motivation is self-efficacy. Bandura [26] defines it as beliefs in one’s capacities to organize and execute courses of action required to produce certain achievements.

As pointed out by Navarro and Quijano [27] “the execution that a person will obtain in his work (as an indicator of his motivated behavior, that is, of the “effort” he has made) will be determined by the intensity or strength of his motives moderated by his perception of self-efficacy” (p. 645). In this sense, the present research analyses the moderated effect of self-efficacy on the relation between the three internal psychological states of employees with intellectual disabilities compared to those with physical disabilities working in Special Employment Centers (SEC) and their motivation.

Following Hackman and Oldham [21,22,23,24] in the present research we propose the following hypotheses:
**Hypothesis** **1.***Increased levels of meaningfulness of work generate increased levels of work motivation*.
**Hypothesis** **2.***Increased levels of experienced responsibility for work generate increased levels of work motivation*.
**Hypothesis** **3.***Increased levels of knowledge of results increased levels of work motivation*.

Additionally, this study aims to analyze if the relationship between employees’ internal psychological states and motivation is moderated by employees’ self-efficacy. Therefore, our research tried to verify if self-efficacy positively interacts with meaningfulness of work, responsibility for work, and knowledge of results to moderate its effect on motivation. In this regard:
**Hypothesis** **4.***Self-efficacy moderates the relationship between meaningfulness of work and work motivation*.
**Hypothesis** **5.***Self-efficacy moderates the relationship between knowledge of results and work motivation*.
**Hypothesis** **6.***Self-efficacy moderates the relationship between responsibility for work and work motivation*.

Figure 1 illustrates the overall conceptual model and hypotheses.

## 2. Method

### 2.1. Participants and Procedures

An accidental sampling strategy was used. Researchers contacted the managers of the centers and invited them to participate. Once accepted, direct support employees administered the survey to participants. It included a cover letter with information about the purpose of the survey, the research ethics protocols, and the survey itself. To facilitate participation and help employees with special needs, in some centers the administration was in group, in a room with computers. For those who needed help, the direct support employee provided a standardized clarification. Participation in the survey was voluntary and strictly confidential.

A total of 187 employees from Special Employment Centers (SEC) in Spain with intellectual (*n* = 71, 38%) and physical (*n* = 116, 62%) disabilities, recognized by the Spanish law [6,9], completed an online survey. Participants had a legally recognized degree of disability ranging between 33% and 82% (Mean = 43.5%, SD = 12.4). A greater number of participants were women (53.5%) and had primary (45.1%) or secondary studies (28.2%). The mean age was 41.81 years (SD = 9.88) and mean tenure 7.36 years (SD = 7.35). Most employees had permanent contracts (60.4%), worked full-time (68.4%), and held production positions (61%). Some of the participants had been previously working in companies in the ordinary market (25.1%).

### 2.2. Measures

Following the International Tests Commission [28] guidelines, scales used in the present research were adapted to the special needs of respondents, avoiding potential effects due to possible difficulties on lecture and comprehension.

Psychological critical states. The subdimension measuring psychological critical states (PCS) of the Psychological Processes Scale (PPS) [29], based on the Job Diagnostic Survey [30] was used. The PCS is composed by 6 items in a 5-points Likert scale, from Strongly disagree to Strongly agree, and evaluates knowledge of results, responsibility for outcomes, and meaningfulness of work.

Some examples of items were “Normally I know whether my work is correct or not” (Knowledge of results), “The results of my work depend to a large extent on my efforts to do my job well” (Responsibility for outcomes), and “I consider that most tasks I have to do in this job are useful and important” (Meaningfulness of work). The scale showed a good internal consistency in the present study (α = 0.725), and its validity via confirmatory factor analysis in Navarro et al. [29].

Motivation. We analyzed the direct motivation from a three-item scale [29], based on the intrinsic motivation of the Job Diagnostic Survey of Hackman & Oldham [30]. This scale has been used in a previous study with people with intellectual disabilities [31]. The internal consistency of the scale measured by Cronbach’s alpha was 0.683 [29] and 0.723 in the present study. Its criterion validity, proven through its correlation with intrinsic work motivation scale developed by Warr et al. [32] was 0.63 [29].

Self-efficacy. The subdimension measuring self-efficacy (two items) of the Psychological Processes Scale (PPS) [29], was used. An example of item was “In general I believe that I am capable of managing my work”. The internal consistency of the scale was moderate in the present study (α = 0.618).

### 2.3. Data Analyses

Initial analyses of group differences (employees with intellectual disabilities and with physical disabilities) were executed by chi-squared and *t*-test. Secondly, we examined correlation matrices to test bivariate relations across all variables. Finally, moderation effect of self-efficacy was conducted using the PROCESS macro created by Hayes [33] in SPSS 23. Based on the pick-a-point approximation, graphical computational tools were also used to further explore the interactions of the predictor and moderator variables [33].

## 3. Results

### 3.1. Descritive Analyses

In relation to socio-demographic variables, non-significant differences were found on gender distribution (χ^2^ = 1.437, *p* = 0.231). Contrarily, differences were found related to age (t = 4.595, *p* < 0.001), tenure (t = −11.092, *p* < 0.001), and contract (full-time/part-time) (χ^2^ = 11.702, *p* = 0.001). In general terms, employees with intellectual disabilities were younger, with more time in company, and with a higher percentage of full-time contracts (Table 1).

Participants showed medium-high levels in all analyzed variables (scores below 4 in a 5-points Likert scale), being the highest scores for meaningfulness of work (M = 4.16; SD = 0.65) and motivation (M = 4.16; SD = 0.69). Comparing both groups, some differences were observed between them on self-efficacy, meaningfulness of work and motivation. In general terms, employees with intellectual disabilities perceived lower levels of self-efficacy and motivation, and higher levels of meaningfulness of work than employees with physical disabilities (Table 2).

### 3.2. Associations between Predictor Variables and Motivation

Considering the global sample, all predictor variables correlated highly and significative with motivation, being responsibility for outcomes and meaningfulness of work those with the highest correlation coefficients. Analyzing separately both groups, motivation did not correlate with knowledge of results on employees with intellectual disabilities. The proposed moderator variable, self-efficacy, correlated with all other variables in both groups and in the global sample (Table 3).

### 3.3. Self-Efficacy Moderation Analysis

The moderation effect of self-efficacy on the relationship between predictor variables and motivation was analyzed on the global sample and by groups (Table 4). On the global sample, self-efficacy moderated the relationship between responsibility for outcomes and meaningfulness of work and motivation but did not moderate the relationship with knowledge of results.

Analyzing both groups separately, the effect of self-efficacy is different depending on the kind of disability of employees. On employees with intellectual disabilities, self-efficacy moderates the relationship between responsibility for outcomes and meaningfulness of work with motivation. Knowledge of results have neither direct nor moderated effect on motivation among this group of employees. Contrarily, among employees with physical disabilities only the relationship between knowledge of results and motivation is moderated by self-efficacy. On this group of employees, responsibility for outcomes and meaningfulness of work have a direct effect on motivation, not moderated by self-efficacy.

Graphical computational allowed to further explore the interactions of the predictors and moderator variables. Specifically, on employees with intellectual disabilities and low levels of self-efficacy there existed a positive relationship between responsibility for outcomes, meaningfulness of work and motivation. That relationship was not confirmed for employees with medium or high levels of self-efficacy (Figure 2).

Contrarily, among employees with physical disabilities and medium or high levels of self-efficacy the knowledge of results had a positive relationship with motivation, but not on employees with low levels of self-efficacy (Figure 3).

## 4. Discussion

The main objective of the present research was to analyze the moderated effect of self-efficacy on the relation between the three internal psychological states of employees with intellectual and physical disabilities working in Special Employment Centers (SECs) and their motivation. Although the model of Hackman and Oldham [22,23,24,30] is nowadays widely studied on different kinds of employees and organizations, to the best of our knowledge, no studies have been conducted exploring the model fit in people with intellectual and physical disabilities.

Additionally, our research adds to the growing literature on the topic of work motivation on employees with disabilities the moderator effect of self-efficacy. Since Bandura’s original work [26] the effect of self-efficacy on motivation has been widely studied in educational [34] and organizational research [27,35,36], but again there are no studies in the context of employees with disabilities. Specifically, we attend their job motivation, as American Psychological Association [37] have shown that more motivated employees are those that feel more valued by their employer, feel treated fairly and trust their employer, and receive sufficient mental health resources from their employers.

Our results have shown that SEC employees with intellectual and physical disabilities had medium to high levels of psychological critical states, self-efficacy and motivation, having those employees with intellectual disabilities lower levels of self-efficacy and motivation and higher levels of meaningfulness of work, compared with their colleagues with physical disabilities.

In accordance with the Job Characteristics Model [22,23,24,30], all three psychological critical states correlated positively with motivation on employees with physical disabilities. This result was not confirmed for employees with intellectual disabilities. In this case, knowledge of results had no relationship with motivation.

In relation with the moderating effect of self-efficacy, we found differences depending on the kind of disability of employees. On employees with intellectual disabilities, their levels of self-efficacy moderated the effect of responsibility for outcomes and meaningfulness of work on motivation. Specifically, we observed an effect of both critical psychological states on motivation only among employees with low levels of self-efficacy. In this sense, it is important to point out that, to increase responsibility for outcomes on employees with low levels of self-efficacy managers should design their jobs providing them with higher levels of autonomy [21,30]. As recent studies have shown interaction effects among autonomy and self-efficacy [38] to explain performance, future research should analyze these relationships more deeply.

On the other hand, to increase meaningfulness of work on employees with low levels of self-efficacy, jobs should be re-designed increasing skills variety, identity and significance [22,30]. Some studies have shown that while skills identity and significance directly predict motivation, this relation is mediated by enjoyment in the case of skills variety [39], but considering that, as stated by Warr [40], skills variety have an additional decrement effect on well-being, following a curvilinear pattern. For this reason, we consider that managers should assure that employees with disabilities enjoy with the skills variety they need to play on in their work.

In the case of employees with physical disabilities, and medium and high levels of self-efficacy, the knowledge of results is a predictor of motivation. To achieve suitable levels of knowledge, it is important that managers provide feedback about the results of the work to employees [22,30]. Our research highlights the importance of the interaction between self-efficacy and awareness of work results. Additionally, we consider that future research should analyze the effect of managers’ feedback on this relation, especially due to the situational and contingent nature of self-efficacy perceptions [24], to have more information about the antecedents of motivation and stablish guidelines for intervention.

Finally, for employees with physical disabilities it is important to note the direct effect of responsibility for outcomes and meaningfulness of work on motivation, independently of their levels of self-efficacy. As both variables are associated with the job enrichment, managers should guarantee that employees have autonomy to develop their work, and jobs that allow them to put into practice varied skills, with identity and significance [22,30]. It implies that employees with disabilities should be empowered to make choices and exercise control in their working lives [41].

The present research has some limitations related to the organizational and national context, sample characteristics, and methodology. Firstly, this research has been conducted in some Special Employment Centers in Spain using an accidental sampling strategy. For this reason, our results couldn’t be generalized to all employees with disabilities working on Spanish SECs or on the ordinary labor market. It is necessary to further analyze the effect of self-efficacy on the relationship between psychological critical states and motivation in different work environments and countries, considering the effect of law regulations and cultures. Related to the characteristics of the jobs developed by participants, in the present research we couldn’t get to assure the anonymity of participants. Some results could be explained if we analyzed the job descriptions, and additionally assure more effective guidelines for intervention.

Secondly, and related to sample characteristics, we can mention size, types and subtypes of disabilities included, as participants self-report their kind of disability as physical or intellectual, and degree of disability of participants. We consider that the number of participants need to be increased in future researches, as well as include employees with other kinds of disabilities, as sensorial or organic. Finally, as we used self-evaluation questionnaires, it could be a potential problem of source bias. Future research should include other methods, as qualitative and official sources of information and a mixed-method approach, to avoid the risk of common-method variance [42].

## 5. Conclusions

In this research, the hypotheses based on the theoretical foundations of the Hackman and Oldham model [22,30] have been partially confirmed, although differences have been observed depending on the type of disability of the employees. Specifically, in employees with intellectual disabilities, their levels of self-efficacy moderated the effect of responsibility for outcomes and meaningfulness of work on motivation. In the case of employees with physical disabilities, the knowledge about results is a predictor of motivation when they had medium or high levels of self-efficacy. Additionally, in this group, responsibility for outcomes and meaningfulness of work had a direct effect on motivation, independently of their levels of self-efficacy. Employees with disabilities should be empowered to make choices and exercise control in their working lives. To do so, human resources managers should promote adapted and contingent policies and strategies to improve their motivation, and consequently their wellbeing, taking into consideration the workforce diversity.

## Figures and Tables

**Figure 1 ijerph-16-01447-f001:**
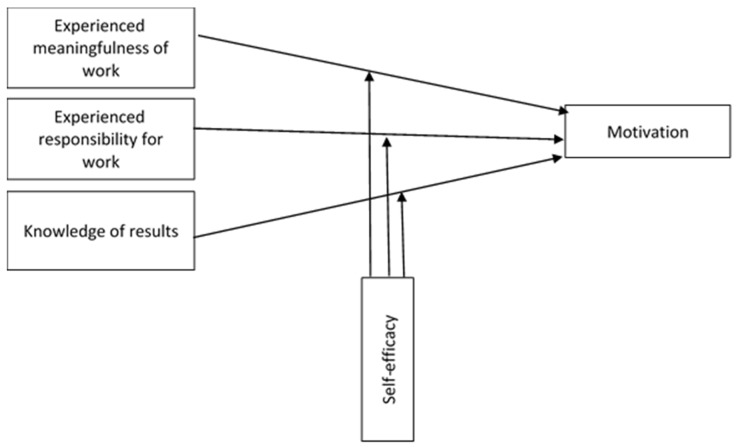
Conceptual model and hypotheses of impact of the three internal psychological states on motivation: the moderating effect of self-efficacy.

**Figure 2 ijerph-16-01447-f002:**
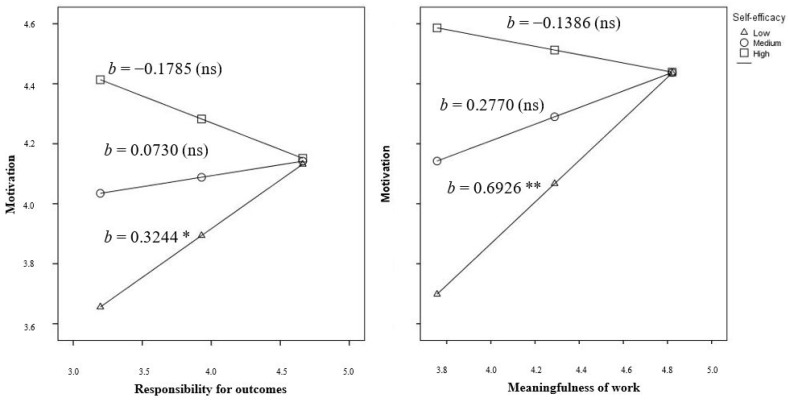
Moderator effect of self-efficacy on the relationship between responsibility for outcomes, meaningfulness of work, and motivation on employees with intellectual disabilities (Note: ** *p* < 0.001, * *p* < 0.01).

**Figure 3 ijerph-16-01447-f003:**
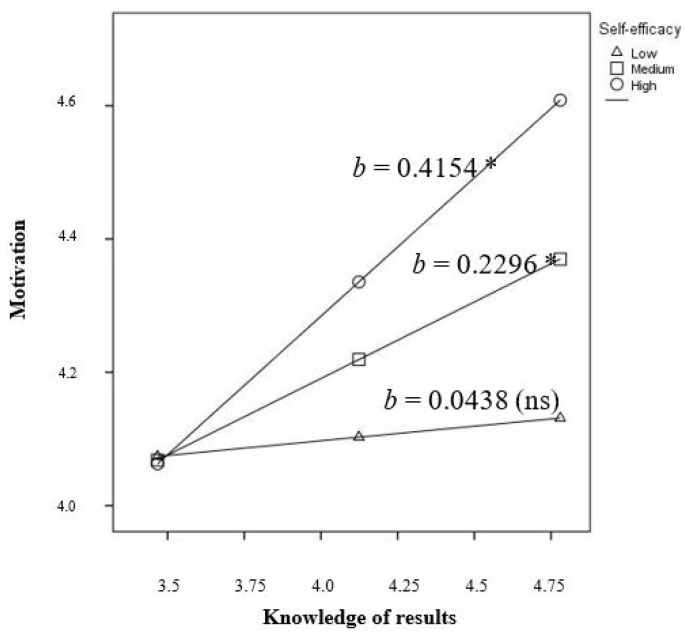
Moderator effect of self-efficacy on the relationship between knowledge of results and motivation on employees with physical disabilities (Note: ** *p* < 0.001, * *p* < 0.01).

**Table 1 ijerph-16-01447-t001:** Sample description (by groups).

Socio-Demographic Variables		N	%	χ^2^ (*p*)
Gender distribution (women)	Physical	66	56.9%	1.437 (0.231)
	Intellectual	34	47.9%	
Full-time contract	Physical	60	60.7%	11.702 (0.001)
	Intellectual	68	84.5%	
		**N**	**Mean (SD)**	**t (*p*)**
Age	Physical	110	44.36 (9.43)	4.595 (<0.001)
	Intellectual	70	37.79 (9.255)	
Tenure	Physical	110	3.37 (3.62)	−11.092 (<0.001)
	Intellectual	67	13.91 (7.24)	

**Table 2 ijerph-16-01447-t002:** Mean comparisons between groups.

Variables		N	Mean	SD	t (*p*)
Self-efficacy	Global sample	186	4.0000	0.69555	2.273 (0.025)
	Physical	115	4.0957	0.61020	
	Intellectual	71	3.8451	0.79549	
Knowledge of results	Global sample	184	4.0598	0.67485	1.633 (0.104)
	Physical	113	4.1239	0.65646	
	Intellectual	71	3.9577	0.69563	
Responsibility for outcomes	Global sample	185	4.0486	0.73189	1.756 (0.081)
	Physical	114	4.1228	0.72430	
	Intellectual	71	3.9296	0.73337	
Meaningfulness of work	Global sample	185	4.1622	0.64747	−2.118 (0.036)
	Physical	114	4.0833	0.70056	
	Intellectual	71	4.2887	0.53226	
Motivation	Global sample	187	4.1604	0.69332	2.359 (0.019)
	Physical	116	4.2529	0.66175	
	Intellectual	71	4.0094	0.72149	

**Table 3 ijerph-16-01447-t003:** Correlation coefficients between variables.

Sample	Dimensions	Motivation	Knowledge of Results	Responsibility for Outcomes	Meaningfulness of Work
Global sample *n* = 187	Knowledge of results	0.265 **			
	Responsibility for outcomes	0.491 **	0.563 **		
	Meaningfulness of work	0.489 **	0.218 **	0.410 **	
	Self-efficacy	0.408 **	0.548 **	0.493 **	0.353 **
Employees with intellectual disabilities *n* = 71	Knowledge of results	0.134			
	Responsibility for outcomes	0.267 *	0.512 **		
	Meaningfulness of work	0.346 **	0.207	0.208	
	Self-efficacy	0.310 **	0.614 **	0.434 **	0.369 **
Employees with physical disabilities *n* = 116	Knowledge of results	0.340 **			
	Responsibility for outcomes	0.627 **	0.586 **		
	Meaningfulness of work	0.634 **	0.259 **	0.555 **	
	Self-efficacy	0.463 **	0.481 **	0.529 **	0.425 **

** *p* < 0.001, * *p* < 0.01.

**Table 4 ijerph-16-01447-t004:** Analyses of the moderator effect of self-efficacy.

Sample	Direct Relationship	b_3_	SE	t	95% CI
Global sample (*n* = 187)	Knowledge of results—Motivation	0.0869	0.0830	1.0472	(−0.0769; 0.2507)
	Responsibility for outcomes—Motivation	−0.1378 *	0.0520	−2.6474 *	(−0.2405; −0.0351)
	Meaningfulness of work—Motivation	−0.1582 *	0.0530	−2.9845 *	(−0.2628; −0.0536)
Intellectual (*n* = 71)	Knowledge of results—Motivation	−0.1054	0.1312	−0.8030	(−0.3672; 0.1565)
	Responsibility for outcomes—Motivation	−0.3161 *	0.1156	−2.7335 *	(−0.5470; −0.0853)
	Meaningfulness of work—Motivation	−0.5225 **	0.1496	−3.4918 **	(−0.8211; −0.2238)
Physical (*n* = 116)	Knowledge of results—Motivation	0.3476 **	0.1227	2.8335 **	(0.1045; −0.5907)
	Responsibility for outcomes—Motivation	−0.0996	0.0532	−1.8725	(−0.2051; 0.0058)
	Meaningfulness of work—Motivation	−0.0880	0.0530	−1.6624	(−0.1930; 0.0169)

** *p* < 0.001, * *p* < 0.01.

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
