# Peer review of "Future Work and Disability: Promoting Job Motivation in Special Employment Centers in Spain"

_ijerph, 2019, doi:10.3390/ijerph16081447_

Round 1

Reviewer 1 Report

It is a very interesting job..

I hope that the authors can soon present a new work by replicating it in other samples of employees without disabilities to compare their results.

Author Response

POINT 1. It is a very interesting job. I hope that the authors can soon present a new work by replicating it in other samples of employees without disabilities to compare their results.

RESPONSE 1. Thank you so much for your comment! As we indicated in Discussion section, it is also of our interest to increase the number of participants in future researches, as well as to include employees with other kinds of disabilities, as sensorial or organic (lines 318-322)

Reviewer 2 Report

Very interesting work - I enjoyed reading.

Abstract:

- it should be mentioned that the study was conduced in Spain. Maybe it makes sense to mention it alreay in the title

- You mention that there are different results regarding the kind of disability but present only results those for intellectually disabled. Short results concerning physically disabled to be added.

Introduction:

First Paragraph: Very important work to be mentioned Frey and Osborne (2013).

- I missed a Definition of disability - how is legally accepted disability defined in Spain? To be mentioned in the Introduction

- Some Information about SECs in Spain to be added - how many exist? Are there Special Centers specialiced on different Kind of disabilities? How many People are "employed" there? Do the disabled receive a monthly income which is high enough to live? How many disabled working in SECs find employment in the ordinary working market. Very important Information for international Readers.

- Please add some statistics on disabled People in Spain: how many disabled exist? And how many with different kind of disabilities? How many are (un)employed? How is their education Level?

Mehod: (l. 111ff.)

- how did you choose the Managers of the SECs? Did you contact all SECs in Spain? From where did you get the adresses?

- Please add Information on representativity.

- Please add description what type of tasks the disabled generally conduct at SECs (maybe in introduction).

- please define what Kind both intellectual and physical disability - what types of disabilities are summed up in These two groups?

- Can you add Information like: when did the persons get disabled, do they have a qualification - and if yes which degree, where they already employed in ordinary labour market before they got into the SEOs?

Results

under every table add Information about the size of Data and name the data

Author Response

POINT 1. Very interesting work - I enjoyed reading.

RESPONSE 1. Thank you so much for your comment!

POINT 2. Abstract:

POINT 2.1. it should be mentioned that the study was conducted in Spain. Maybe it makes sense to mention it already in the title

RESPONSE 2.1. We include this information on Title and Abstract (lines 3, 12).

POINT 2.2. You mention that there are different results regarding the kind of disability but present only results those for intellectually disabled. Short results concerning physically disabled to be added.

RESPONSE 2.2. On Abstract, we present the results for intellectually disabled employees: “In the case of employees with physical disabilities, the knowledge of results is a predictor of motivation when they had medium or high levels of self-efficacy. Additionally, in this group, responsibility for outcomes and meaningfulness of work had a direct effect on motivation, independently of their levels of self-efficacy” (lines 16-21).

POINT 3. Introduction:

POINT 3.1. First Paragraph: Very important work to be mentioned Frey and Osborne (2013).

RESPONSE 3.1. Thank you for your suggestion. This work is interesting and relevant, and we included it as a reference (Reference [4]).

POINT 3.2.I missed a Definition of disability - how is legally accepted disability defined in Spain? To be mentioned in the Introduction

RESPONSE 3.2. The legal definition on disability in Spain is included in the Royal Decree 1971/1999, December 23rd, Procedure for the recognition, declaration and qualification of the degree of disability (Spanish Law). We included it as a reference on Participants and procedures section (lines 34-37, reference [6]).

POINT 3.3. Some Information about SECs in Spain to be added - how many exist? Are there Special Centers specialized on different Kind of disabilities? How many People are "employed" there? Do the disabled receive a monthly income which is high enough to live? How many disabled working in SECs find employment in the ordinary working market. Very important Information for international Readers. Please add some statistics on disabled People in Spain: how many disabled exist? And how many with different kind of disabilities? How many are (un)employed? How is their education Level?

RESPONSE 3.3. Following your recommendations, we included more information about SECs and about situation of people with disability in Spain from the General Report about people with disabilities on the labor market and the Spanish Public Service of Employment (Introduction section) (lines 43-48 and 57-65).

POINT 4. Mehod: (l. 111ff.)

POINT 4.1. how did you choose the Managers of the SECs? Did you contact all SECs in Spain? From where did you get the addresses?

RESPONSE 4.1. On Participants and procedures section we included that an accidental sampling strategy was used. We contacted some centers we had relationships and they put in contact with other SECS (snowball) (line 129).

POINT 4.2. Please add Information on representativity.

RESPONSE 4.2. We agree. Our results couldn’t be generalized to all employees with disabilities working on Spanish SECs or on the ordinary labor market. For this reason, on Discussion section we included that it is necessary to further analyze the effect of self-efficacy on the relationship (lines 306-322).

POINT 4.3. Please add description what type of tasks the disabled generally conduct at SECs (maybe in introduction).

RESPONSE 4.3. Following your recommendations, we included this information on Introduction (lines 64-65)

POINT 4.4. please define what Kind both intellectual and physical disability - what types of disabilities are summed up in These two groups?

RESPONSE 4.4. In order to guarantee the anonymity of participants, the authors did not collected information about the sub-types included. In this sense, as we indicated in Discussion section: “Secondly, and related to sample characteristics, we can mention size, types and subtypes of disabilities included, as participants self-report their kind of disability as physical or intellectual, and degree of disability of participants. We consider that the number of participants need to be increased in future researches, as well as include employees with other kinds of disabilities, as sensorial or organic.” (lines 318-322).

POINT 4.5.Can you add Information like: when did the persons get disabled, do they have a qualification - and if yes which degree, where they already employed in ordinary labour market before they got into the SEOs?

RESPONSE 4.5. Unfortunately, in order to guarantee the anonymity of participants we do not have information related to your first question, but we included information about qualifications (level of studies) (lines 140-141). Additionally, following your suggestion, we added information about the percentage of participants that came from the ordinary market to SEC (lines 143-144).

POINT 5. Results

POINT 5.1. under every table add Information about the size of Data and name the data

RESPONSE 5.1. Tables 1 (line 183) and 2 (line 192) included the sample size in each variable and the name of data in all columns. We added information about sample size in Table 3 (line 199). Table 4 is adapted to APA and journal style (line 215).

Reviewer 3 Report

According to the conceptual framework on the used research design, it stated that; 

    A 187 employees with intellectual and physical disabilities completed the Internal Motivation         scale, 12 the psychological critical states (PCS) and the self-efficacy sub-scale of the Psychological Processes 13 Scale (PPS). 

THAT MEANS PERSONS WITH MENTAL DISABILITIES ARE EITHER INVISIBLE  OR EXCLUDED AS POSSIBLE BENEFICIARIES REGARDLESS OF CONSTITUTING THE THIRD  CATEGORY OF PERSONS WITH DISABILITIES AT LEAST ACCORDING TO THE CONVENTION ON RIGHTS OF PERSONS WITH DISABILITIES. 

Despite of admitting  the above exclusion in the trends of the study and perhaps explain the rationale for its justification, the paper is instead going further to make assert conclusions about employees with disabilities in a much generic context. Generic in a manner that seemingly creates a contravercial debates as to whether all the above three categories of persons disabilities were part of the participants which the study subjected completing the (PCS) and (PPS). That means this paper would be more interesting if the authors give readers some clarity as to reasons influencing or informing their choices of the representative sample of the selected categories of disability. 

Secondly the study seems to be referring to participants with disabilities in the Global North, considering the likelihood of socioeconomic variances in working life and context between developed and developing economies, such a contextual setting would make the paper more insightful and enriching to the readers. 

Author Response

POINT 1. According to the conceptual framework on the used research design, it stated that;

A 187 employees with intellectual and physical disabilities completed the Internal Motivation         scale, 12 the psychological critical states (PCS) and the self-efficacy sub-scale of the Psychological Processes 13 Scale (PPS).

THAT MEANS PERSONS WITH MENTAL DISABILITIES ARE EITHER INVISIBLE OR EXCLUDED AS POSSIBLE BENEFICIARIES REGARDLESS OF CONSTITUTING THE THIRD CATEGORY OF PERSONS WITH DISABILITIES AT LEAST ACCORDING TO THE CONVENTION ON RIGHTS OF PERSONS WITH DISABILITIES.

Despite of admitting the above exclusion in the trends of the study and perhaps explain the rationale for its justification, the paper is instead going further to make assert conclusions about employees with disabilities in a much generic context. Generic in a manner that seemingly creates a controversial debate as to whether all the above three categories of persons disabilities were part of the participants which the study subjected completing the (PCS) and (PPS). That means this paper would be more interesting if the authors give readers some clarity as to reasons influencing or informing their choices of the representative sample of the selected categories of disability.

RESPONSE 1. We agree with the reviewer. Nevertheless, participants self-reported their kind and level of disability, and the authors decided not to collect information about sub-types of disability in order to guarantee the anonymity of participants. For these reasons, our sample have some limitations related to representativeness that we added in Discussion section (lines 308-322).

POINT 2. Secondly the study seems to be referring to participants with disabilities in the Global North, considering the likelihood of socioeconomic variances in working life and context between developed and developing economies, such a contextual setting would make the paper more insightful and enriching to the readers.

RESPONSE 2. Following reviewers’ suggestions, we clarified that our research is focused in Spanish SECs on Title and Abstract section (lines 3, 12). For this reason, as we pointed out on Discussion section our results couldn’t be generalized to other work environments and countries, considering the effect of law regulations, economic conditions and cultures (lines 309-314).

Reviewer 4 Report

The paper contributes to the research on employment for persons with disabilities and perceptions related their job satisfaction and motivation - an under-researched area. 

While scientifically thorough, the paper could be strengthened by fleshing out both the limitations section and the conclusion. 

Limitations: The paper's consideration of only those persons with disabilities who are employed at the Special Employment Centers (SECs), as noted, is a serious limitation as this is not reflective of those in the mainstream labor market - which is the ideal promoted. A definition of the SECs should be included - is this subsidized employment, "work creation', or, in fact, meaningful constructive work in a safe, supportive environment?  The limitations section should also note why further disaggregation of the sample is not included - physical disabilities, for example, covers a wide spectrum of diverse disabilities - mobility impairments, visual impairments, hearing impairments, etc., and those with intellectual disabilities covers a wide-range of intellectual functioning from mild to severe. There is also no reference to overlap - that is, those who might have more than one type of disabilities. One might assume that those with varying degrees of impairment might report differently. 

Conclusion:  This section should reiterate all the major findings and conclusions presented in the paper in summary form. The current conclusion does not do this.   

Author Response

POINT 1. Limitations. The paper's consideration of only those persons with disabilities who are employed at the Special Employment Centers (SECs), as noted, is a serious limitation as this is not reflective of those in the mainstream labor market - which is the ideal promoted.

RESPONSE 1. We agree. We clarified this on Discussion section (lines 308-322).

POINT 2. A definition of the SECs should be included - is this subsidized employment, "work creation', or, in fact, meaningful constructive work in a safe, supportive environment? 

RESPONSE 2. As we express in Introduction section the SECs are defined as organizations that should have as a social objective the inclusion of people with disabilities and can be created directly by the public administrations or by natural or legal persons who meet the appropriate civil requirements. Following the current regulations, the SECs are required to have "the necessary instruments to be able to have structures, systems, processes and forms of organization that guarantee their competitiveness, not only to maintain themselves regularly in the market, but also to increase their position in it”. Following reviewers’ suggestions, we added additional information related to SECs characteristics (lines 57-65).

POINT 3. The limitations section should also note why further disaggregation of the sample is not included - physical disabilities, for example, covers a wide spectrum of diverse disabilities - mobility impairments, visual impairments, hearing impairments, etc., and those with intellectual disabilities covers a wide-range of intellectual functioning from mild to severe. There is also no reference to overlap - that is, those who might have more than one type of disabilities. One might assume that those with varying degrees of impairment might report differently.

RESPONSE 3. We agree with the reviewer but, unfortunately, in order to guarantee the anonymity of participants we did not collected information about the sub-types included, as we indicated in Discussion section as a limitation (lines 318-322).

POINT 4. Conclusion. This section should reiterate all the major findings and conclusions presented in the paper in summary form. The current conclusion does not do this.

RESPONSE 4. As stated in Instructions for authors, “this section is mandatory, and should provide readers with a brief summary of the main conclusions”. Following reviewer suggestion, we added to our conclusions a summary of results in this section (lines 329-335).

Round 2

Reviewer 3 Report

You have made changes in lines 302-322 dealing with how selection of the different categories of disabilities were made. 

The limited respsentativeness of the sample could explained clearly 

Generally speaking the addition of Spanish Emploment centres in the title address the second issue that had been raised in the earlier review